# Structural Characterization of Micronized Lignocellulose Date Pits as Affected by Water Sonication Followed by Alcoholic Fractionations

**DOI:** 10.3390/ijms26146644

**Published:** 2025-07-11

**Authors:** Khalid Al-Harrasi, Nasser Al-Habsi, Mohamed A. Al-Kindi, Linghong Shi, Hafiz A. R. Suleria, Muthupandian Ashokkumar, Mohammad Shafiur Rahman

**Affiliations:** 1Department of Food Science and Nutrition, College of Agricultural and Marine Sciences, P.O. Box 34-123, Al-Khod 123, Muscat, Oman; k.s.al99@hotmail.com (K.A.-H.); habsin@squ.edu.om (N.A.-H.); 2Department of Pathology, College of Medicine and Health Sciences, Sultan Qaboos University, P.O. Box 35-123, Al-Khod 123, Muscat, Oman; alkindi2@squ.edu.om; 3School of Agriculture, Food and Ecosystem Sciences, Faculty of Science, The University of Melbourne, Parkville, VIC 3010, Australia; 4Sonochemistry Group, School of Chemistry, The University of Melbourne, Parkville, VIC 3010, Australia; masho@unimelb.edu.au

**Keywords:** nanoparticles, sonicated, alcoholic fractionation, particle size, glass transition, crystal, amorphous, date pits, SEM, TEM

## Abstract

Date pits are considered waste, and micronized date pit powder could be developed for use in foods and bio-products. In this study, micronized date pit powders were extracted by alcoholic sedimentation after ultrasound treatment. The control was considered untreated, i.e., without sonication. Six micronized fractions (i.e., three from control and three from treated) were prepared by three stages of alcoholic sedimentation. In the case of untreated date pit powder, the average particle size of the fractionated date pit powder (i.e., residue) from three stages of alcoholic sedimentation varied from 89 to 164 µm, while ultrasonic treatment showed variation from 39 to 65 µm. The average particle size of the supernatant fractions of untreated date pit powder varied from 22 to 63 µm, while ultrasonic treatment showed variation from 18 to 44 µm. Ultrasound treatment produced smaller particles. In all cases, Scanning Electron Microscopy (SEM) showed that supernatant fractions contained lumped particles compared to the residue fractions. Transmission Electron Microscopy (TEM) showed the presence of nanoparticles in all extracted fractions. Two glass transitions were observed in all fractions except for the residue from the first sedimentation stage. In addition, higher levels of degradation in the fractionated date pits could be achieved by ultrasonic treatment, as is evident from the Fourier Transform Infrared (FTIR) analysis.

## 1. Introduction

Date pits are a by-product of date fruit processing factories that can be used as a low-cost additive in food products to enhance fiber content and bioactive compounds [1]. They are used in beverages, desserts, condiments, meat products, and bakery products [2]. Particle surface area increased with the decrease in particle size and affected hydration and oil absorption properties. The hydrolysis and the particle size of the added date pits increased the healthy functional polyphenols and changed the flavor and texture of muffins incorporated with date pits [3]. Zamzam et al. [4] studied the effect of particle size at 150 and 300 μm on chocolate at different ratios. They observed the highest homogeneity when date pit powder of 150 μm was added at a ratio of 1:9. However, improved taste was observed at the ratios of 1:9 and 3:7 with a particle size of 150 μm. Characteristics of particles (i.e., size) are important when date pits are utilized in food products [5]. Particles of micron size can be incorporated as dietary fiber in fiber-fortified foods [6]. The optimal ratios of added date pits depend on the types of food, particle size, and treatments [2]. These could allow us to incorporate date pits at a higher level in food products. In the case of coconut powder, Raghavendra et al. [7] observed that the reduction in particle size from 1127 to 550 μm resulted in increased hydration properties. This could be due to an increase in surface area and surface porosity as well as structural modification. However, hydration properties decreased with the decrease in particle size when the particle size decreased from 550 μm, which showed the opposite trend when compared to the higher particle size. Therefore, the particle size of fortified powder in foods is an important property that needs to be known before identifying its uses.

It is important to understand the physicochemical properties of fibers when they are used in food products to enhance sensory, structural, and health functionality, and how they provide other health functionalities in the digestive tract in the presence of diversified microorganisms and digestive enzymes. Al-Khalili et al. [8,9] extracted crystalline and amorphous date pit fractions after alkali digestion. They pointed out that a crystalline fraction of date pits could enhance the formation of the desired crust, and an amorphous fraction could enhance the softness of crumbs in bread. Crystalline fibers are commonly used in food products for retaining the stability of ice cream, fat substitutes or replacers, bulking agents, encapsulated wall materials for protection, and reinforcement in edible films. They are insoluble in nature and stable in a colloidal system, thus exhibiting a fat-like mouthfeel. They formed a particle gel network and an inert molecule, which filled the gaps of a tight meat fiber network and did not disturb the protein network. This functionality improved the mechanical properties of formulated meat products. The crystalline fiber was used to replace fats in emulsions, baked products, frozen desserts, mayonnaise, gravies, and sauces. This was due to its low solubility and provided a fat-mimicking texture. Micro- and nanocrystalline components increased the specific surface area, water-holding capacity, swelling capacity, and oil-holding capacity. Similarly, amorphous cellulose provided a fat substitute in fermented sausages and provided a gel-like texture [10]. Plasticized powders are closer to the soluble fibers due to their ability to form a gel-like structure.

The health functionality of fibers depends on their solubility, crystallinity, and amorphous or plastic characteristics. Soluble fibers showed more effectiveness in lowering diabetes, heart disease, cholesterol levels, and obesity [11,12], while showing less effectiveness in preventing colorectal cancer [13]. Crystalline cellulose showed the ability to eliminate contaminants by adsorbing carcinogens since it showed a low degree of digestion [14]. Crystalline components are commonly unaffected by gut microorganisms, and their low fermentability improves fecal bulking with smooth flow [15]. De Maesschalck et al. [16] used amorphous cellulose as a feed-supplemented diet. They observed that amorphous or plasticized cellulose did not show inertness since it altered the gut microbial environment. Digestive processes and microbial fermentation depend on the degree of solubility and viscous mass [17]. In addition, damaged fibers could have the ability to form gels with interlinked networks.

Sakr et al. [18] studied the physicochemical, structural, and functional properties of Moringa leaf powder with three particle sizes (i.e., 1.65, 3.76, and 8.53 μm). Different particle sizes were created by a home grinder and ball mill. They observed that specific surface areas, bulk and trap densities, and water solubility indices increased with the decrease in particle size. Moreover, the angle of repose, water holding capacity and oil holding capacity decreased with the increase in particle size. Scanning Electron Microscopy (SEM) showed particle surface roughness decreased with the decrease in particle size. Solids’ melting temperature decreased with the decrease in particle size. With decreasing crystal size, FTIR showed a lack of formation of new functional groups; however, crystallinity pattern remained the same. In addition, FTIR absorbance of typical peaks decreased with increasing particle size.

Ke et al. [5] prepared edible homology plant powder by ball mill grinding with different grinding times. The average particle sizes of five fractions were 10.44, 10.21, 10.06, 9.83 and 9.96 μm, respectively. With the decrease in particle sizes, they observed increased zeta-potential, coarser particles, as observed using SEM, and increased fat and glucose absorption. However, water-holding capacity, oil-holding capacity, water-swelling capacity, total phenols, total flavonoid and ion exchange capacity did not show any changes. Particles sized 10.06 μm presented the greatest change in the above categories, while powder with particles sized 9.83 μm showed the highest ion exchange capacity.

At present, most reports are based on the basic grinding process of whole date pits, and their use in food products. Other treatments, such as physical (i.e., ultrasound and high-pressure steam cooking), chemical (i.e., alcohol, acids and alkali), microbial, and enzymatic hydrolysis need to be applied to improve the properties of date pits. Al-Mawali et al. [19] produced three types of extracted fibers from defatted date pit powder using alkali treatment at 30 °C. They extracted supernatant and residue fractions by maintaining different levels of pH. Residue date pit powder was produced by alkaline-sodium chloride and potassium hydroxide treatments [20]. Four types of whole date pit powder were produced by water soaking, ethanol pretreatment, microwave heating and pressure cooker cooking [21]. Ultrasound decreased particle size during treatment [22]. Awad et al. [23] reviewed the applications of ultrasound in modifying food products and found that it was relatively cheap, simple and saved energy. Mostafa et al. [24] prepared whole date pit powder using a seed grinder with particles passing through a 250 μm sieve. It was then further treated with hydrochloric acid and ultrasound. The produced powder contained nano-size particles within 50 to 150 nm depending on the treatment. They characterized the treated powder by SEM, FTIR, and phenolic, flavonoid, and antioxidant capacity. A variety of food components, such as oil, protein, polysaccharides, and bioactive components, were extracted from plant and animal sources using ultrasound [25]. Aliyu and Hepher [26] observed that ultrasound at 80 W and 38 kHz fragmented cellulosic biomaterial to glucose and other chemicals.

Considering the applications of fibers in foods and other bioproducts, it is important to know the characteristics of particle (i.e., size and shape), amorphous or crystalline nature, and degree of damage of the fibers. This could determine their performance in achieving desired characteristics. Velsaco et al. found that the degree of fiber damage affected the debonding of composite materials [27]. In the literature, chemical treatments are commonly presented to treat date pit powder and only one type of treated powder has been developed. Limited works are presented on natural treatments and extraction of different date pit fractions with varied particle size.

The objectives of this study were to develop different types of date pit powder by water sonication followed by three stages of the alcoholic sedimentation process. Twelve fractionated date pit powders (i.e., six untreated and six ultrasound-treated) were developed. Three stages of alcoholic fractionations were used for the treated and untreated date pits. In addition, particle size and morphology, alongside thermal and functional group characteristics, were measured. The possible utilization of fractionated date pit powder is discussed based on its measured characteristics. The characteristics of each type of fiber were aligned with their possible uses in foods and bioproducts and when these could be beneficial.

## 2. Result and Discussion

### 2.1. Fractionation Mechanisms

The supernatants (OS_1_ and TS_2_) contained the highest levels of lignin compared to other fractions since lignin could easily dissolve in alcohol. Earlier, it was also observed that supernatant fractionations of date pits suspended in alcohol contained a high level of lignin [7,21]. Alcohol as a solvent was used to fractionate date pits since alcohol was reported to separate lignin and holocellulose from lignocellulosic materials [28]. Ethanolic extraction provided a balance between efficiency of lignin extraction and the structure of the resulting lignin [29]. In addition, ethanol was easy to separate from the extracted fractions. Alcohol disrupted the lignin–carbohydrate bonds and suspended in alcohol. The solvent sedimentation process produced different fractions of varied particle sizes. The fractionation was due to density differences of particles. The average size of particles increased with the increase in the fractionation stage, i.e., the first-stage extraction produced smaller particles (Table 1). Ultrasonic treatment further decreased the particle size of date pits, since it created and caused tiny bubbles in a liquid, resulting in expanding and collapsing. This generated a powerful force that can break particles and disrupt lignin–carbohydrate bonds [30,31].

### 2.2. Particle Size Distribution

Table 1 shows the mass average particle size distribution of water-soaked (OWD) and ultrasonic-water-treated (TWD) fractionated whole date pits. The data reveal that most particles in the OWD- and TWD-treated whole date pits were larger, ≥250 μm (i.e., 85.58% and 84.95%, respectively). The smaller size fraction of the treated sample was increased to 1.44% as compared to 0.48% in the case of the untreated sample (Table 2). This indicated that ultrasonic treatment effectively produced smaller particles in the date pits.

Table 1 illustrates the average particle size of the fractionated (i.e., treated and untreated) whole date pits. Average particle sizes of the treated fractions (i.e., residues and supernatants) were smaller as compared to the untreated fractions (*p* < 0.05). The treated supernatant (i.e., TS_1_) showed the smallest average size (i.e., 18 ± 9 μm), while untreated residue (i.e., OR_3_) showed the largest particle size (i.e., 164 ± 106 μm). It was clear that ultrasonic treatment reduced the average particle sizes of all the fractionated date pit powder (Table 1). The average particle sizes of the fractionated date pits at the nano level were measured by TEM and it showed similar trends in the case of micro-size particles, as measured by SEM. In other words, ultrasonic treatment also decreased the average sizes of all fractions (i.e., residues and supernatants). Mostafa et al. [24] observed that ultrasound treatment decreased the particle size of acid-hydrolyzed whole date pit powder.

### 2.3. Scanning Electron Microscopy (SEM)

Figure 1 shows the particle morphology of the extracted date pit fractions (i.e., without sonication). The particle size of the residue fractions increased with the increased extraction stages. This was due to the separation of smaller particles with the supernatant in stage one and stage two. The shapes of the third residue fraction were relatively spherical, while the first and second residue fractions contained some needle-shaped particles. In addition, particles in the first residue were lumped together. However, all water-soaked treated supernatant fractions showed relatively lumped particles.

Figure 2 shows the particle morphology of the ultrasonic-water-treated residue and supernatant fractions. All the treated fractions showed relatively lumped particles except residues TR_2_. The TR_2_ fraction showed relatively fine particles, and these are not agglomerated; however, TR_3_ showed that the particles are intact and separated. In the case of Moringa leaf powder, Sakr et al. [18] observed increased particle roughness with the decrease in size. This may cause the lumping of the particles with a decrease in particle size, as observed in this study. In the case of edible homology plant powder, an increased agglomeration was observed with the decrease in particle size [5]. If the particle size was small, and the specific surface area was large, agglomeration occurred more easily among the particles [32].

### 2.4. Transmission Electron Microscopy (TEM)

TEM microscopy was performed to identify the presence of nano size particles in the water-soaked and ultrasonic-water-treated fractions (Figure 3 and Figure 4). The ultrasonic-water treatment decreased the particle sizes of all fractionated residues and supernatants. Similarly, Mostafa et al. [24] observed that ultrasonic-water treatment decreased the particle size to the nano-level of acid-hydrolyzed whole date pit powder, as measured by Zetasizer. However, they did not visualize the nano-size particles by TEM or AFM.

### 2.5. Differential Scanning Calorimetry (DSC)

Figure 5 shows a typical DSC thermogram of the selected fraction (i.e., TS_1_). It shows two shifts as glass transitions (i.e., G_1_ and G_2_) and one endothermic peak as solids melting (i.e., M). There was an structural building (i.e., H) after solids melting (i.e., M). Glass transition 2 shows an endothermic peak (i.e., E) after the glass transition is evident. In the case of date pits, an endothermic peak was evident after the glass transition [2,19]. Table 3 shows the characteristics of the first and second glass transitions. The first glass transition varied from 126 to 144 °C in the water-soaked and ultrasonic-water-treated fractions. The TR_1_ fraction did not show the first glass transition. This indicates that this TR_1_ fraction resulted in crystalline or higher order, with minimal amorphous components. The specific heat change at the first glass transition (∆*C_p_*)_1_ increased in all ultrasonic-treated fractions (i.e., residues and supernatants). This shows clearly that ultrasonic treatment caused plasticization in the treated fractions. Regarding treated residues, (∆*C_p_*)_1_ increased with the increase in extraction stages, while in the case of supernatants, it decreased. Therefore, treated residues contained more plasticized components, while the treated supernatant contained more ordered molecular fractions. On the other hand, water-soaked supernatants showed a similar decreasing tendency, with the increase in extraction stage. However, in the case of residues, no trend was observed.

A lower trend in the second glass transition temperature was observed in the case of ultrasonic treatment (*p* < 0.05). This could be due to the molecular defects caused by the ultrasound. With regard to residue, the second glass transition temperature decreased with the increase in extraction stage, whereas there was an increasing trend in the case of supernatant. Similar trends were also observed in the case of ultrasonic-water treatment. Treated residue fraction TR_1_ showed the lowest (∆*C_p_*)_2_ (i.e., 64 J/kg °C), while TS_2_ and TS_3_ showed the highest change in the (∆*C_p_*)_2_ (i.e., 1018 and 1109 J/kg °C, respectively). Therefore, TR_1_ was the most crystalline or ordered fraction, while OS_1_, TS_2_, and TS_3_ were the most plasticized fractions. Therefore, TR_1_ was stable in the long term, compared to the other factions; however, OS_1_, TS_2_, and TS_3_ were the least stable due to the high number of plasticized fractions. In relation to the applications, crystalline powder could be used in bio-composite as fillers, and crust formation in food products. It can also be used to stabilize ice cream and make particle gel-like colloidal structure in emulsions, frozen desserts, mayonnaise, gravies and sauce, and exhibits a fat-like mouth feel. These crystalline fractions of date pit powder can be used to eliminate contamination or carcinogens and provide a low degree of digestion [14]. These could be less affected by gut microorganisms and improve fecal bulking with smooth flow in the digestive tract [15]. However, amorphous or plasticized fractions could be good in the formation of soft crumb in bread and gel-like interlinked matrix in food products, such as biscuits, cakes and muffins [21]. Plasticized fractions could be easily fermented in the digestive system and could be formed into a viscous mass. Plasticized fibers are close to soluble fibers since these have the ability to form a viscous mass with an interlinked matrix, and could be beneficial in lowering the incidence of diabetes, heart diseases, high cholesterol, and obesity.

The solid melting temperature varied from 181 to 192 °C, whereas melting enthalpy varied from 50 to 174 J/g (Table 4). It was difficult to find any clear tendency on the solid melting peak as a function of particle size. Sakr et al. [18] observed a decreasing peak temperature with the decrease in particle size of Moringa leaf powder. The decreasing particle size caused the exposure of specific components (e.g., protein and polysaccharides) and this caused the instability of the powder. The lowest solid melting enthalpy (i.e., 50 J/g) was observed for TR_1_ and this fraction was completely different from the observed glass transition (Table 4). The highest enthalpy was observed in the case of OR_1_ (i.e., 174 J/g).

### 2.6. Fourier Transform Infrared (FTIR) Analysis

FTIR spectra of water-soaked (OR_1_, OR_2_, OR_3_ and OS_1_, OS_2_, and OS_3_) and ultrasonic-water-treated samples (OR_1_, OR_2_, OR_3_ and OS_1_, OS_2_, and OS_3_) are shown in Figure 6 and Figure 7. The degree of damage in the functional groups decreased with the increase in extraction stage (Figure 6A,B) of water-soaked fractionated residues. However, in the case of fractionated supernatants, the fraction from the first extraction stage (i.e., OS_1_) showed the lowest damage compared to the fractions from stages two and three (Figure 6C,D). Similar observations were seen in the ultrasonic-water-treated fractions (Figure 7). However, the degree of damage (i.e., absorption intensity) depended on the types of functional groups. In the case of Moringa leaf powder, FTIR spectroscopy showed similar functional groups; however, absorbance of typical peaks decreased with the increasing particle size. This indicated lower structural damage of the functional groups in the case of larger particles [18]. Similarly, Ke et al. [5] also observed that there was no change in the type of functional groups in dietary fibers; however, peak strengths were enhanced. In the case of defatted millet bran with high pressure and high temperature with ultrasonic treatment, there was no appearance of new peaks, indicating that there was no modification in the functional groups; however, ultrasonic-treated rice bran showed the highest absorption intensities [33]. The degree of structural damage of lignocellulosic fibers indicated how fibers could interact in food and the bio-composite matrix. High molecular damage (i.e., structural splitting) in the date pits’ fractionated fibers, as evidenced by FTIR analysis, indicated their ability to form a bonding or structural network when used in foods or bio-composite [21]. The fractions OS_1_, TR_1_, and TS_1_ showed the highest molecular damage as compared to other fractions (Figure 6 and Figure 7). Therefore, these fractions could form gel-like interlinked matrices, whereas less damaged fractions could be used to make colloidal matrices in food products.

In whole date pit powder (OWD), O-H stretching was observed within 3401–3455 cm^−1^, whereas in the case of TWD treatment, these were variations within 3371–3408 cm^−1^. These stretches could be observed from the O-H functional group from alcohol, water, and hydrates. These could also appear from amino compounds and ammonium compounds, however date pits did not contain these compounds [24]. These could be the stretching effects of the backbone of cellulose and hemicellulose [34,35]. In the case of untreated and treated date pits (i.e., hydrolyzed and hydrolyzed-ultrasound), these peaks were observed to be within 3316–3290 cm^−1^, which, in turn, was lower than observed in this study [24]. The OR_1_ and OS_1_ showed the highest intensities (i.e., 0.421 and 0.365 AU), while OR_3_ showed the lowest intensity (i.e., 0.107 AU), indicating minimal damage to the O-H stretching in OR_3_. Ultrasonic-water-treated samples showed higher intensities, except samples TS_2_ and TS_3_ (i.e., 0.773 and 0.570 AU). This indicated less damage to the O-H stretching by ultrasound. The peak of OS_1_ was sharp and split into three as compared to the other samples, which indicated a higher level of damage in the O-H stretching (Figure 6D).

The peaks within 2925–2935 cm^−1^ were due to the C-H stretching, while the peaks within 2861–2858 cm^−1^ were due to the H-C-H stretching vibration. These could be stretching in methyl and methylene group of cellulose [34,35]. C-H stretching existed in all ultrasonic-water-treated and water-soaked samples, whereas H-C-H was absent in the cases of OR_2_, OS_1_, OS_3_, TR_2_, TR_3_, and TS_1_. In all ultrasonic-water-treated samples, these intensities were higher, which indicated a higher damage to these functional groups by ultrasound. In the case of untreated date pits, two intense and sharp peaks at 2910 and 2924 cm^−1^ were observed, while acid-hydrolyzed and hydrolyzed-ultrasound-treated date pits showed one low peak and another wide peak [24].

Weak peaks were obtained between 1864–1871 cm^−1^ and 1833–1836 cm^−1^. All extracted fractions showed absorption bands within 1739–1749 cm^−1^. These are due to the carboxylic (i.e., C=O-OH) or carbonyl (C=O) groups. This is possibly due to ester linkage of carboxylic groups of the ferulic and p-coumaric acids of lignin and/or hemicelluloses or carbonyl group of acetyl and uronic ester groups within the structure of hemicelluloses. Similar peaks were also observed in the whole date pits and residue fibers of alkaline–salt–alkaline treatment. The absorption intensities varied from 0.032 to 0.601 cm^−1^ in the case of samples without ultrasound treatment, while it varied from 0.059 to 0.768 cm^−1^ in the case of ultrasound-treated samples.

The observed absorption peaks within 1625–1648 cm^−1^ were due to the C=C stretching of aromatic skeletal mode, and peaks within 1519–1543 cm^−1^ could be due to the aromatic C=C bending. The peak appearing around 1540 cm^−1^ can also be assigned to the aromatic C=C bending vibration [36]. These could be due to the stretching and bending of the aromatic ring of the lignin. However, peaks within 1442–1612 cm^−1^ could be due to the aromatic stretching of C=C from lignin [34,35].

The peaks 1446–1459 cm^−1^ were due to C-H bending. The peaks were observed 1377–1394 cm^−1^ due to the C-H bending of cellulose and hemicellulose. It was mentioned that it could be within 1371–1427 cm^−1^ in the case of lignocellulosic biomaterials [34,35]. Ultrasonic-water-treated fractionated date pits showed peaks within wave number 1322–1323 cm^−1^ in the case of TR_2_ and TS_1_, while other water-soaked fractionated samples did not show these peaks.

The peaks within 1244–1268 cm^−1^, 1102–1159 cm^−1^ and 1057–1070 cm^−1^ were due to the glycosidic C-O-C bond vibration. The glycosidic bond vibration was observed within cellulose and hemicellulose at peaks within 1000–1244 cm^−1^ [34,35]. In the case of millet bran, a strong absorption was observed at 1124 cm^−1^ due to glycosidic bond vibration and OH deformation vibration [33]. The sample without sonication (i.e., OS_3_) showed a peak at wave number 998 cm^−1^. Only samples without sonication (i.e., OR_1_, OS_2_, and OS_3_) and with sonication (i.e., TR_1_ and TR_3_) indicated peaks between 933 and 943 cm^−1^. Peaks within 869–877 cm^−1^ indicated the existence of mannosidic bonds. A similar peak at 897 cm^−1^ was observed in the case of millet bran [33] and tea polysaccharides [37].

The peaks observed between 810–824 cm^−1^ and 773–779 cm^−1^ were due to C-H bending. Samples with sonication (i.e., TR_1_, TS_2_ and TS_3_) indicated peaks within 712–725 cm^−1^, whereas these peaks were absent in the case of other samples. The sonicated sample (i.e., TR_1_, TR_2_, TR_3_, TS_2_, and TS_3_) showed peaks within 521–554 cm^−1^, while the samples without sonication did not show these peaks. In the case of selected functional groups, stretching and bending are identified. Stretching indicated a change in length of a bond, whereas bending indicated curving, i.e., change in bond angles.

In the literature, crystallinity from FTIR signals was defined considering the ratio of the bands at specific wave numbers based on models [14]. Nada et at. [38] defined FTIR crystallinity index as the ratio of 3340 cm^−1^/1337 cm^−1^ (i.e., ratio of O-H stretching and C-H bending). Considering the original extracted samples, the values of the ratios were 2.83, 2.04, 5.63, 1.79, 3.16, and 5.74 for OR_1_, OR_2_, OR_3_, OS^1^, OS_2_, OS^3^, respectively. This indicated that OS_3_ had the highest crystalline characteristic and OS_1_ had the the highest amorphous characteristic and OS_1_ provided similarity with the DSC crystallinity, as shown in Table 3 (i.e., lowest and highest ∆*C_p_* changed at glass transition). However, the order of crystalline characteristics did not follow the same order as in Table 3. Considering the ultrasound extracted samples, the values of the ratios were 2.88, 1.75, 1.63, 2.40, 2.00, and 2.38 for TR_1_, TR_2_, TR_3_, TS_1_, TS_2_, and TS_3_, respectively. This indicated that TR_1_ had the highest crystalline characteristic and TR_3_ had the highest amorphous characteristic. This was similar to the highest DSC crystalline nature for TR_1_, but different in the lowest value for TS_3_, as shown in Table 3. Therefore, FTIR crystallinity did not exactly represent structural behavior as observed by DSC. This could be due to the possibility of additional relaxation at the glass transition in addition to the ∆*C_p_* change due to the fraction of crystalline component in the sample [39].

## 3. Materials and Methods

### 3.1. Raw Materials

Dates of the Medjool variety (i.e., tamr stage) (5 kg) were obtained from a local market in Melbourne (Victoria, Australia) for this experiment. Flesh and seeds were separated from whole date pits. Date pits were dried in a fume hood for a week and then reduced to powder with an 800 g grinder (Laobenhang, model 400Y, Yongkang, Zhejiang, China). The flesh and seed samples were then stored at −20 °C until further analysis.

### 3.2. Extraction of Different Fractions

The whole date pit powder was soaked in distilled water at a ratio of 1:10 and dried at 60 °C for 24 h (OWD). Different fractions were extracted using absolute ethanol. Three main stages of extraction protocol are shown in Figure 8 and extracted powders are marked as residues (OR_1_: first residue, OR_2_: second residue, OR_3_: third residue) and supernatants (OS_1_: first supernatant, OS_2_: second supernatant, OS_3_: third supernatant). The extraction process involved soaking at 25 °C for 24 h with continuous stirring. The treated sample was centrifuged using 8000 rpm at 4 °C for 15 min (Hettich Refrigerated Centrifuge ROTINA380R Tuttlingen Baden Württemberg, Germany). Supernatant and residue were pre-frozen at −80 °C for 12 h and then freeze-dried at −60 °C for 72 h (Labconco Benchtop Freeze Dryer, Kansas City, MO, USA).

Another batch of date pit powder was soaked in distilled water with a ratio of 1:10 (i.e., 4 g sample in 40 mL water in a 50 mL tube). The mixture in the beaker was treated by sonication for 5 min at 25 °C using an ultrasonic probe (Digital Sonifier 450, Branson Ultrasonics, Brookfield, CT, USA) operating at 20 kHz and 40% amplitude (nominal power: 400 W) (TWD). The sonication probe was directly dipped into the mixture and then the sonication process was turned on. The treated powder was dried similarly to the one mentioned earlier. Different fractions were extracted in ethanol as mentioned earlier and these were marked as residues (TR_1_: first residue, TR_2_: second residue, TR_3_: third residue) and supernatants (TS_1_: first supernatant, TS_2_: second supernatant, TS_3_: third supernatant). All fractions were equilibrated at an atmospheric relative humidity of 11.1% (i.e., water activity: 0.111) and then had their properties measured.

### 3.3. Measurement of Particle Size Distribution

A vibratory sieve shaker (Retsch GmbH, Haan, Germany) was used to determine the particle size distribution of the untreated and sonicated extracted date pit powders. Sieves with different sizes (i.e., 1000, 500, 250, 125, 90 and 63 µm, respectively) were assembled one above another according to their size. Sieves were tightly fixed on the shaker and the shaker was set for 10 min at 50 rpm. Sample powder (10 g) was placed in the middle of the first sieve (i.e., 1000 μm) and then run in the shaker. The fractions were weighed and used to calculate the mass of particles in each sieve. Mass average particle sizes were calculated.

### 3.4. Morphological Analysis by Scanning Electron Microscopy (SEM)

The morphological structure of the treated and untreated date pit fractions was scanned using Scanning Electron Microscopy (SEM) (FESEM: JEOL JSM-5600 LV, Tokyo, Japan) operated at 20 kV with a working distance of 20 mm. Samples were prepared according to the method used by Suresh et al. [40]. A sample was placed on the top of an SEM stub using double-sided carbon adhesive tape and then coated with a thin layer of gold (i.e., 10–20 nm) to prevent charging during measurement and increase the conductivity of a sample using Jeol Smart Coater DIL-29030SCTR, Tokyo, Japan. Stapes were exposed to a 0.2 Torr vacuum pressure using a sputter coater (SPI-module, Santa Clara, CA, USA). Digital micrographs were captured at various magnifications and saved on a computer.

### 3.5. Morphological Analysis by Transmission Electron Microscopy (TEM)

Morphological structures of the nano-size present in the extracted date pit powder were measured using Transmission Electron Microscope (TEM) (JEOL, JEM-5600 LV, Tokyo, Japan) operated at 80 kV. Samples were soaked in ethanol and a drop from the supernatant was placed on the TEM 300-mesh formvar carbon-coated grids. Digital micrographs were recorded at different magnifications and saved on a computer [41].

### 3.6. Thermal Analysis by Differential Scanning Calorimetry (DSC)

The thermal behavior of different extracted samples (i.e., fibers) was analyzed by Differential Scanning Calorimetry (DSC) (DSC Q20, TA Instrument, New Castle, DE, USA) [29]. The sample (i.e., approximately 3–5 mg) was placed in a Tzero hermetic DSC aluminum pan and sealed with a lid. A blank empty pan was considered as reference. Nitrogen gas flow (i.e., 50 mL min^−1^) was used during DSC measurement. Samples were run considering the protocol used by Al-Harrasi et.al. [42]. A shift in the heat flow curve was referred to as glass transition; however, the endothermic peaks represented oil melting or solids melting. Data were analyzed by Universal Analysis 2000 DSC Q20 program (v5.5.3).

### 3.7. Functional Group Measurement by Fourier Transform Infrared (FTIR) Analysis

Extracted date pit powder and potassium bromide were mixed thoroughly in a mortar–pestle at a ratio of 1:100 (i.e., 0.02 g sample and 2 g KBr). A hydraulic press (Atlas Manual Hydraulic Press, GS25011, Specac, London, UK) was used to prepare a tablet of 13 mm diameter by compressing the mixture under 10 tons (i.e., 739 MPa) of hydraulic pressure. Samples were analyzed via Bruker FTIR (Bruker, Ettlingen, Germany) spectrometer [20]. The dry condition of the instrument was maintained by using nitrogen gas flow. A background spectrum was verified at 20 °C before running the experimental sample. Fifteen replicates were measured in triplicated reading for 5 tablets. FTIR data were recorded from 4000 to 400 cm^−1^ wave number with 32 scans at a resolution of 4 cm^−1^. Collected data was presented as means ± standard deviation (SD).

### 3.8. Statistical Analysis

Triplicate analyses were applied for each extracted date pit fraction. The data were presented as mean ± standard deviation (SD). One-way analysis of variance (ANOVA) was conducted to test the significance of difference between samples, while Tukey tests were performed to determine significant differences between means (IBM SPSS Statistics 23, v23.0.0.0). The significance was considered when *p* < 0.05.

## 4. Conclusions

Twelve types of micronized date pit powder were developed by ultrasonic-treated and water-soaked date pits followed by three-stage alcoholic sedimentation. The use of ultrasound treatment with alcoholic sedimentation produced date pit powder fractions with different particle sizes possessing different characteristics, i.e., amorphous, crystalline, and degrees of molecular damage. The average particle size varied from 18 to 164 µm and nanoparticles were also observed in all fractions. However, the mass fraction of nanoparticles was not measured. In the residues and supernatants from control (i.e., water-soaked) and ultrasonic-treated, the average particle size increased with the increase in the stages of alcoholic sedimentation. The developed micronized date pit powder possessed varying degrees of amorphous or crystalline (i.e., higher order) characteristics with varied molecular damage. This study indicated that in addition to the particle size, types of fractionated particles also affected the molecular and structural characteristics. In the future, it is important to apply date pit powder in food products to determine its improvement in functionality.

## Figures and Tables

**Figure 1 ijms-26-06644-f001:**
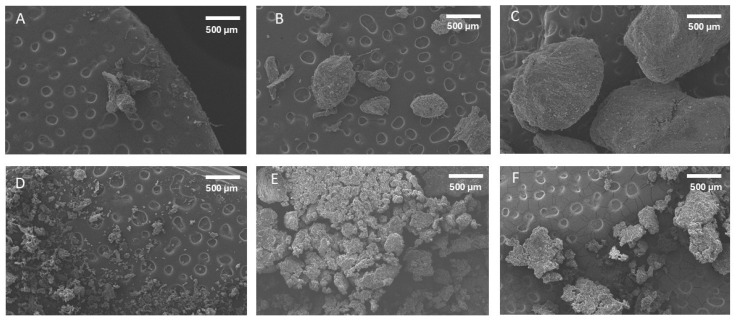
Scanning Electron Micrograph (SEM) of water-soaked (OWD) date pits without ultrasonic treatment. (**A**) (OR_1_): first residue, (**B**) (OR_2_): second residue, (**C**) (OR_3_): third residue, (**D**) (OS_1_): first supernatant, (**E**) (OS_2_): second supernatant, (**F**) (OS_3_): third supernatant.

**Figure 2 ijms-26-06644-f002:**
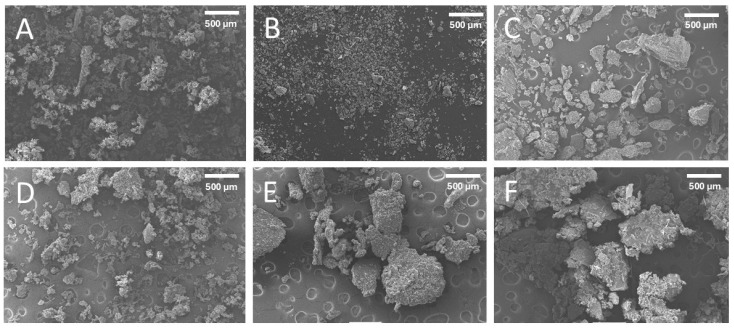
Scanning Electron Micrograph (SEM) of ultrasonic-water-treated (TWD). (**A**) (TR_1_): first residue, (**B**) (TR_2_): second residue, (**C**) (TR_3_): third residue, (**D**) (TS_1_): first supernatant, (**E**) (TS_2_): second supernatant, (**F**) (TS_3_): third supernatant.

**Figure 3 ijms-26-06644-f003:**
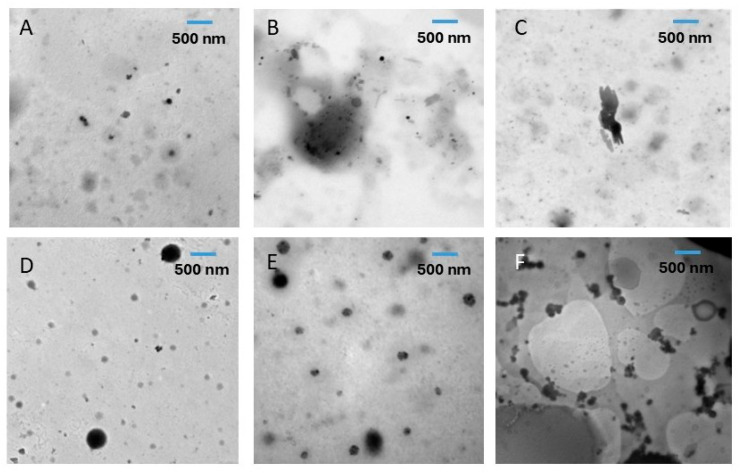
Transmission Electron Microscopy (TEM) of water-soaked (OWD) date pits without ultrasonic treatment. (**A**) (OR_1_): first residue, (**B**) (OR_2_): second residue, (**C**) (OR_3_): third residue, (**D**) (OS_1_): first supernatant, (**E**) (OS_2_): second supernatant, (**F**) (OS_3_): third supernatant.

**Figure 4 ijms-26-06644-f004:**
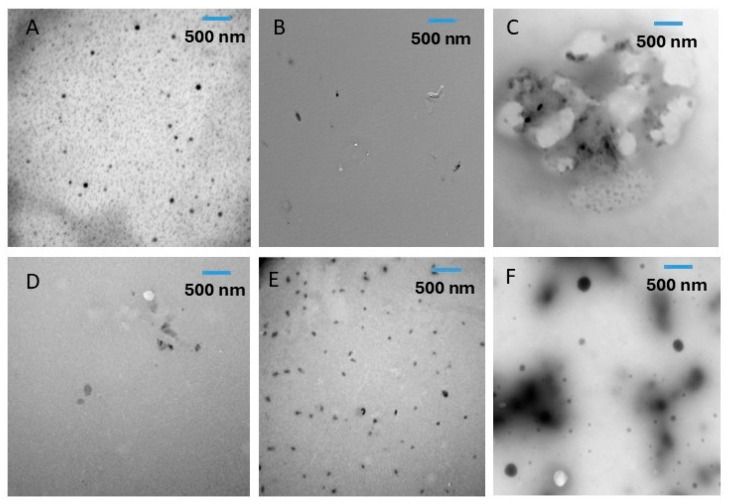
Transmission Electron Microscopy (TEM) of ultrasonic-water treatment (TWD). (**A**) (TR_1_): first residue, (**B**) (TR_2_): second residue, (**C**) (TR_3_): third residue, (**D**) (TS_1_): first supernatant, (**E**) (TS_2_): second supernatant, (**F**) (TS_3_): third supernatant.

**Figure 5 ijms-26-06644-f005:**
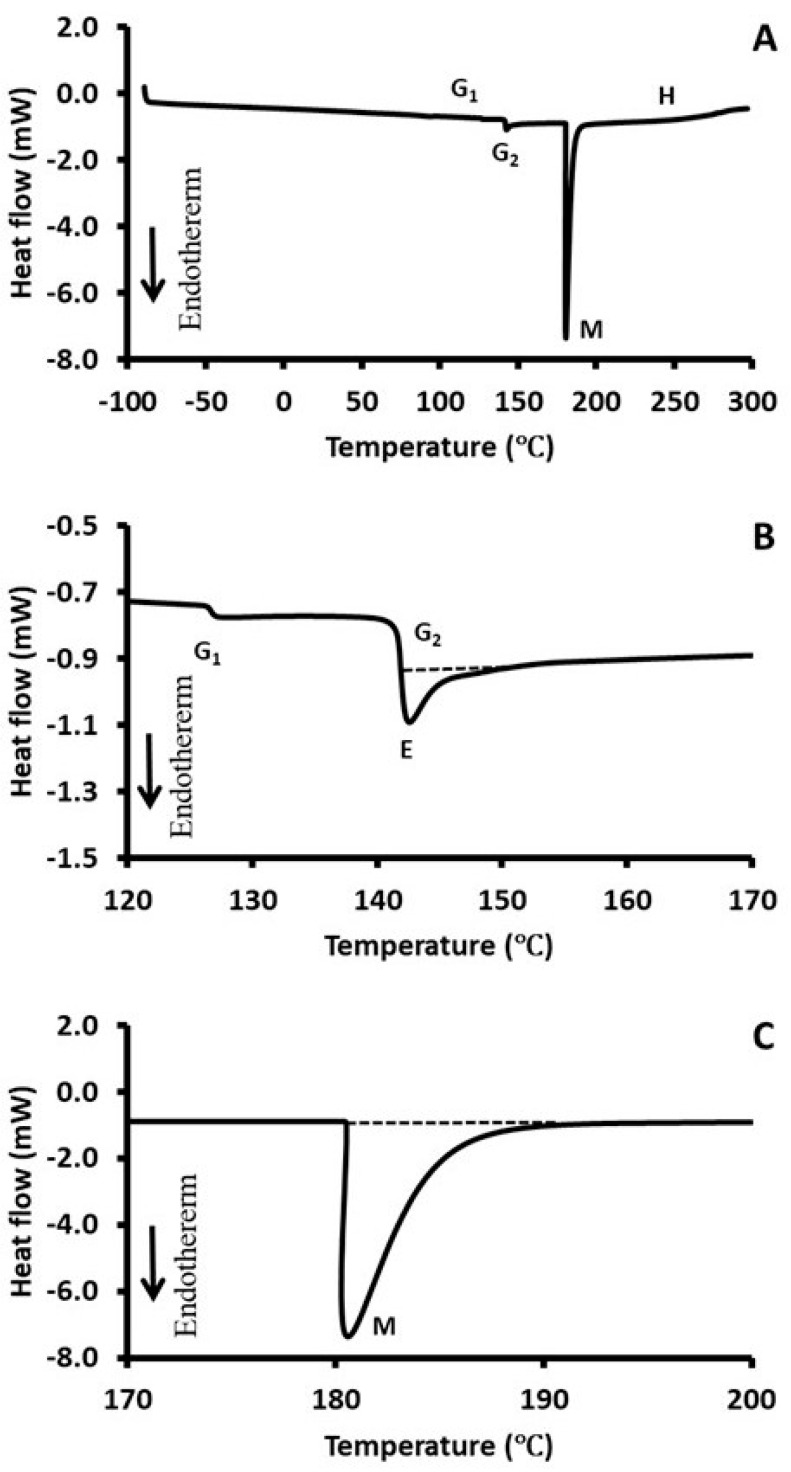
A typical heat DSC flow curve of extracted date pits with ultrasonic-water treatment (i.e., TS_1_). (**A**): Complete thermogram, (**B**): rescaled to clearly visualize first and second glass transitions, (**C**): rescaled to show solids melting.

**Figure 6 ijms-26-06644-f006:**
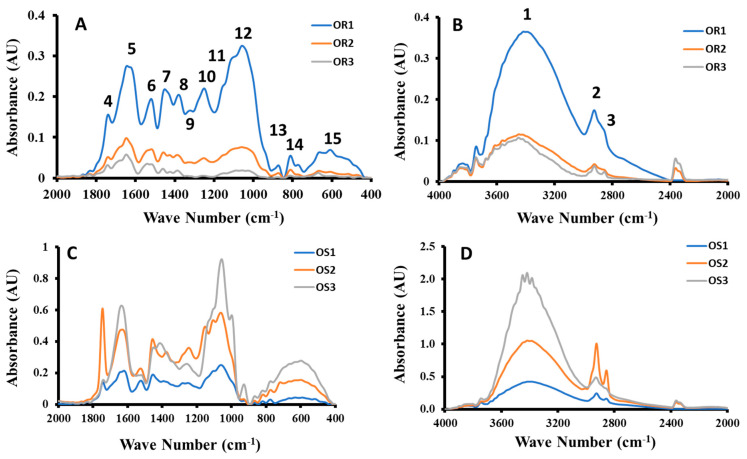
FTIR spectra of selected date pit fractions with water soaking without ultrasonic treatment followed by alcoholic sedimentation of date pits (OWD). (**A**): Residue with 2000 to 200 cm^−1^, (**B**): residue within 4000 to 2000 cm^−1^, (**C**): supernatant with 2000 to 400 cm^−1^, (**D**): supernatant with 4000 to 2000 cm^−1^ (1: O-H stretching, 2: C-H stretching, 3: H-C-H stretching, 4: C=O-OH or C=O, 5 and 6: C=C, 7, 8 and 9: C-H bending, 10, 11 and 12: C-O-C vibration, 13 and 14: C-H bending, 15: C-OH and C-C stretching).

**Figure 7 ijms-26-06644-f007:**
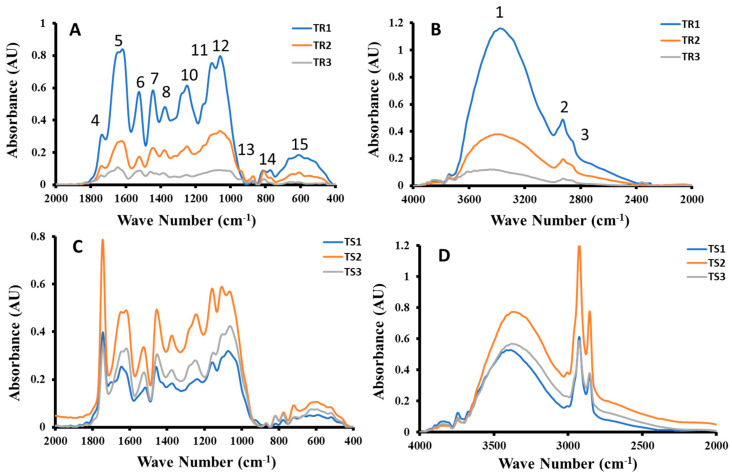
FTIR spectra of selected date pit fractions with ultrasonic-water treatment followed by alcoholic sedimentation of date pits (TWD). (**A**): Residue with 2000 to 200 cm^−1^, (**B**): residue within 4000 to 2000 cm^−1^, (**C**): supernatant with 2000 to 400 cm^−1^, (**D**): supernatant with 4000 to 2000 cm^−1^ (1: O-H stretching, 2: C-H stretching, 3: H-C-H stretching, 4: C=O-OH or C=O, 5 and 6: C=C, 7 and 8: C-H bending, 10, 11 and 12: C-O-C vibration, 13 and 14: C-H bending, 15: C-OH and C-C stretching).

**Figure 8 ijms-26-06644-f008:**
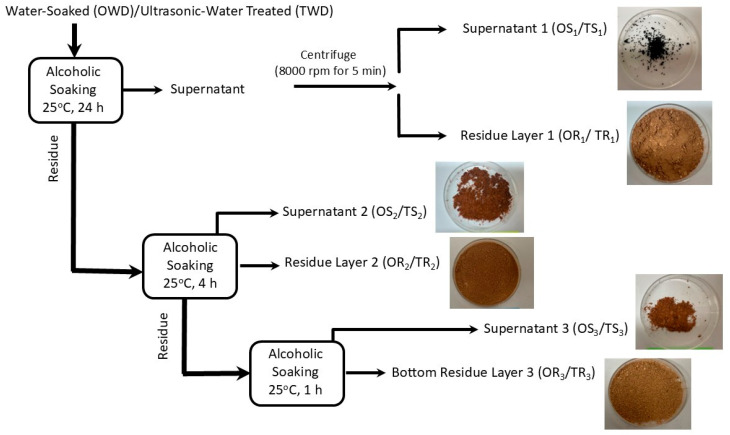
Flow diagram of the water-soaked or ultrasonic-water-treated whole date pit powder followed by alcoholic sedimentation: water-soaked (OWD), OR_1_: first residue, OR_2_: second residue, OR_3_: third residue, OS_1_: first supernatant, OS_2_: second supernatant, OS_3_: third supernatant. Ultrasonic-water-treated whole date pit powder followed by alcoholic sedimentation (TWD). TR_1_: first residue, TR_2_: second residue, TR_3_: third residue, TS_1_: first supernatant, TS_2_: second supernatant, TS_3_: third supernatant (photos included for treated date pits).

**Table 1 ijms-26-06644-t001:** Particle size in the range of macro- and nano-level in the alcoholic fractionated whole and ultrasound-treated date pits.

		Particles Size in the Micro-Range (µm)		Particles Size in the Nano-Range (nm)	
Fiber	Fraction	Size Range	Average Size	Size Range	Average Size
OWD	OR_1_	15–137	89 ± 27 ^c^	44–98	73 ± 15 ^b^
	OR_2_	21–354	98 ± 49 ^c^	60–134	98 ± 20 ^b^
	OR_3_	38–494	164 ± 106 ^b^	71–1335	235 ± 114 ^c^
	OS_1_	5–57	22 ± 19 ^a^	48–76	66 ± 8 ^b^
	OS_2_	7–122	59 ± 20 ^a^	55–86	73 ± 11 ^b^
	OS_3_	28–219	63 ± 25 ^a^	78–196	139 ± 52 ^a^
			*p* < 0.05	*p* < 0.05	*p* < 0.05
TWD	TR_1_	8–79	39 ± 20 ^c^	13–85	42 ± 14 ^a^
	TR_2_	25–95	44 ± 15 ^c^	25–80	52 ± 9 ^d^
	TR_3_	34–150	65 ± 25 ^b^	194–234	91 ± 45 ^c^
	TS_1_	6–54	18 ± 9 ^a^	13–41	27 ± 7 ^a^
	TS_2_	11–78	38 ± 19 ^c^	15–57	30 ± 10 ^a^
	TS_3_	14–143	44 ± 34 ^c^	14–119	49 ± 28 ^d^
			*p* < 0.05	*p* < 0.05	*p* < 0.05

Note: OWD: water-soaked, TWD: ultrasonic-water-treated. Data are presented as mean ± SD. Water-soaked (Control): OR_1_: first residue after centrifuge, OR_2_: second residue, OR_3_: third residue, OS_1_: first supernatant after centrifuge, OS_2_: second supernatant, OS_3_: third supernatant. Ultrasonic-water-treated (TWD): TR_1_: first residue after centrifuge, TR_2_: second residue, TR_3_: third residue, TS_1_: first supernatant after centrifuge, TS_2_: second supernatant, TS_3_: third supernatant. Different letters in a column indicate significant difference (*p* < 0.05).

**Table 2 ijms-26-06644-t002:** Average particle size of the water-soaked and ultrasonic-alcoholic-treated whole date pits.

Sample	Fiber	Mass (%)
OWD	≥250	85.58
	≥125	13.94
	≥63	0.48
TWD	≥250	84.95
	≥125	13.61
	≥63	1.44

Note: Data are presented as mean%. OWD: water-soaked whole date pits, TWD: ultrasonic-water-treated whole date pits.

**Table 3 ijms-26-06644-t003:** First and second glass transitions of different water-soaked and ultrasonic-water-treated samples followed by alcoholic fractionation by sedimentation.

		First Glass Transition				Second Glass Transition			
Sample	Fraction	*T_gi_ *(°C)	*T_gp_* (°C)	*T_ge_* (°C)	(∆*C_p_*)_1_ (J/kg °C)	*T_gi_ *(°C)	*T_gp_* (°C)	*T_ge_* (°C)	(∆*C_p_*)_2_ J/kg (°C)
OWD	OR_1_	144 ± 9 ^f^	144 ± 9 ^f^	144 ± 9 ^f^	105 ± 7 ^b^	158 ± 5 ^d^	158 ± 5 ^d^	158 ± 5 ^d^	636 ± 8 ^c^
	OR_2_	129 ± 1 ^bc^	129 ± 1 ^bc^	129 ± 1 ^bc^	232 ± 1 ^e^	146 ± 3 ^abc^	146 ± 3 ^abc^	146 ± 3 ^abc^	696 ± 3 ^d^
	OR_3_	133 ± 1 ^cd^	133 ± 1 ^cd^	134 ± 1 ^cd^	205 ± 8 ^e^	147 ± 2 ^abc^	147 ± 2 ^abc^	147 ± 2 ^abc^	705 ± 1 ^e^
	OS_1_	129 ± 2 ^bc^	129 ± 2 ^bc^	129 ± 2 ^bc^	224 ± 6 ^f^	144 ± 3 ^ab^	144 ± 3 ^ab^	145 ± 3 ^ab^	928 ± 1 ^i^
	OS_2_	139 ± 2 ^ef^	140 ± 1 ^ef^	140 ± 2 ^ef^	113 ± 8 ^b^	152 ± 8 ^bcd^	152 ± 8 ^bcd^	152 ± 8 ^bcd^	773 ± 2 ^h^
	OS_3_	139 ± 2 ^ef^	139 ± 2 ^ef^	140 ± 2 ^ef^	112 ± 3 ^b^	153 ± 4 ^cd^	153 ± 4 ^cd^	153 ± 4 ^cd^	477 ± 4 ^b^

TWD	TR_1_	N	N	N	N	153 ± 9 ^cd^	153 ± 9 ^cd^	153 ± 9 ^cd^	64 ± 1 ^a^
	TR_2_	130 ± 1 ^bc^	130 ± 1 ^bc^	130 ± 1 ^bc^	178 ± 6 ^c^	145 ± 1 ^a^	145 ± 1 ^a^	145 ± 1 ^a^	736 ± 2 ^f^
	TR_3_	131 ± 1 ^bcd^	131 ± 1 ^bc^	131 ± 1 ^bcd^	270 ± 1 ^g^	145 ± 1 ^abc^	145 ± 1 ^abc^	145 ± 1 ^abc^	762 ± 8 ^g^
	TS_1_	126 ± 1 ^b^	127 ± 1 ^b^	128 ± 1 ^b^	242 ± 8 ^f^	141 ± 1 ^a^	141 ± 1 ^a^	141 ± 1 ^a^	926 ± 3 ^i^
	TS_2_	136 ± 1 ^de^	136 ± 1 ^de^	137 ± 1 ^de^	195 ± 2 ^d^	152 ± 2 ^bcd^	152 ± 2 ^bcd^	152 ± 2 ^bcd^	1018 ± 4 ^j^
	TS_3_	141 ± 2 ^f^	141 ± 2 ^ef^	142 ± 2 ^f^	170 ± 9 ^c^	150 ± 3 ^bcd^	150 ± 3 ^bcd^	151 ± 4 ^bcd^	1109 ± 8 ^k^

Note: Data are presented as mean ± SD. OWD: water-soaked (control): OR_1_: first residue after centrifuge, OR_2_: second residue, OR_3_: third residue, OS_1_: first supernatant after centrifuge, OS_2_: second supernatant, OS_3_: third supernatant; TWD: ultrasonic-water-treated: TR_1_: first residue after centrifuge, TR_2_: second residue, TR_3_: third residue, TS_1_: first supernatant after centrifuge, TS_2_: second supernatant, TS_3_: third supernatant, N: not detected. Different letters in a column indicate significant difference (*p* < 0.05).

**Table 4 ijms-26-06644-t004:** Solid melting of different fractionated date pit powder.

	Solid Melting—Decomposition				
Sample	Fraction	*T_mi_* (°C)	*T_mp_* (°C)	*T_me_* (°C)	∆*H* (J/g)
OWD	OR_1_	190 ± 5 ^abc^	192 ± 5 ^bc^	219 ± 9 ^e^	174 ± 8 ^h^
	OR_2_	192 ± 1 ^bc^	193 ± 1 ^bc^	201 ± 8 ^abc^	87 ± 4 ^d^
	OR_3_	192 ± 1 ^bc^	193 ± 1 ^bc^	206 ± 7 ^bcd^	73 ± 1 ^c^
	OS_1_	187 ± 8 ^abc^	188 ± 7 ^abc^	196 ± 6 ^abc^	111 ± 3 ^f^
	OS_2_	189 ± 3 ^abc^	193 ± 2 ^bc^	215 ± 4 ^de^	81 ± 5 ^cd^
	OS_3_	185 ± 2 ^ab^	186 ± 1 ^ab^	192 ± 1 ^a^	137 ± 3 ^g^

TWD	TR_1_	184 ± 9 ^ab^	186 ± 8 ^ab^	199 ± 3 ^abc^	50 ± 5 ^a^
	TR_2_	181 ± 8 ^a^	182 ± 8 ^a^	196 ± 1 ^ab^	100 ± 3 ^e^
	TR_3_	197 ± 5 ^c^	197 ± 5 ^c^	207 ± 4 ^cd^	86 ± 4 ^d^
	TS_1_	186 ± 1 ^ab^	187 ± 1 ^ab^	199 ± 2 ^abc^	105 ± 1 ^ef^
	TS_2_	191 ± 6 ^abc^	192 ± 5 ^bc^	202 ± 8 ^abc^	64 ± 3 ^b^
	TS_3_	189 ± 3 ^abc^	191 ± 4 ^abc^	198 ± 5 ^abc^	100 ± 9 ^e^

Note: Data are presented as mean ± SD. OWD: water-soaked (control): OR_1_: first residue after centrifuge, OR_2_: second residue, OR_3_: third residue, OS_1_: first supernatant after centrifuge, OS_2_: second supernatant, OS_3_: third supernatant; TWD: ultrasonic-water-treated: TR_1_: first residue after centrifuge, TR_2_: second residue, TR_3_: third residue, TS_1_: first supernatant after centrifuge, TS_2_: second supernatant, TS_3_: third supernatant. Different letters in a column indicate significant difference (*p* < 0.05).

## Data Availability

The additional data supporting the manuscript are available from the corresponding author upon request.

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
