# Peer review of "Structural Characterization of Micronized Lignocellulose Date Pits as Affected by Water Sonication Followed by Alcoholic Fractionations"

_ijms, 2025, doi:10.3390/ijms26146644_

Round 1
Reviewer 1 Report
Comments and Suggestions for Authors
- It provide a performance comparison with other researchers in this field, to better highlight the cutting-edge nature and advantages of this research.
- It is suggested that the author further explain the reaction mechanism.
- It is suggested that the author further elaborate on the prospects and application value of the research conducted in this paper.
- It is suggested to further discuss the long-term stability of composite materials and their performance under various environmental conditions.
- It is suggested to standardize the reference format.
- What is the mass fraction of each of the six micronized components obtained from the processing of jujube seed powder?
- It is recommended to conduct a surface area test on the material.
Author Response
Referee 1
- It provides a performance comparison with other researchers in this field, to better highlight the cutting-edge nature and advantages of this research.
Response: Discussions on other methods are now included on the novelty of this research.
- It is suggested that the author further explain the reaction mechanism.
Response: Included the mechanisms of alcoholic extraction, sedimentation process and ultrasound treatment.
- It is suggested that the author further elaborate on the prospects and application value of the research conducted in this paper.
Response: Values and applications of the extracted date-pits fractions are now included.
- It is suggested to further discuss the long-term stability of composite materials and their performance under various environmental conditions.
Response: The long-term stability determination was outside the scope of the paper. However, based on the glass transition (i.e. crystallinity and plasticized fractions) the stability of the fractionated powder was discussed.
- It is suggested to standardize the reference format.
Response: It is now revised.
- What is the mass fraction of each of the six micronized components obtained from the processing of jujube seed powder?
Response: The mass fractions of two micronized powders, i.e. original and ultrasound treated before alcoholic fractionation were reported in Table 1. However, mass fractions of six micronized components did not record. Yes, it would be good if it was recorded.
- It is recommended to conduct a surface area test on the material.
Response: It would be good to measure the surface area of the extracted fibers. However, it was not targeted in this current study. However, from SEM and TEM images the surface area could be estimated from the size and shape of the particles. We hope to include in our future works.
Reviewer 2 Report
Comments and Suggestions for Authors
The manuscript (ijms-3628162) presents the use of ultrasounds (US) for the processing of date-pits and product characterization. As expected, the US afforded smaller particle size solid residues and has some impact on the characteristics of the resulting residues and extracts. In its current form the manuscript requires some rearrangement and improvement of the presentation:
- Introduction must be improved to clearly highlight previous studies correlated with date-pits and ultrasounds assisted processing.
- The novelty of the study is not clearly presented.
- DSC analysis – Tg attribution must be clearly highlighted and discussed relative to chemical structure and the functional groups involved in value modification.
- Please indicate functional groups on FTIR spectra, please correctly attribute stretching or bending etc. Please improve the discussion.
- What are the uses of the ethanol extracts?
- Conclusions presentation must be improved to highlight the novel aspects of the study.
- Methodology related to US parameters is minimal
Author Response
Referee 2
- Introduction must be improved to clearly highlight previous studies correlated with date-pits and ultrasounds assisted processing.
Response: It is included in the introduction section
- The novelty of the study is not clearly presented.
Response: It is now clearly included.
- DSC analysis – Tg attribution must be clearly highlighted and discussed relative to chemical structure and the functional groups involved in value modification.
Response: DSC and FTIR results are now analyzed and compared based on the literature.
- Please indicate functional groups on FTIR spectra, please correctly attribute stretching or bending etc. Please improve the discussion.
Response: Function groups are indicated as number in the spectrum and the numbers are indicated in the figure captions. The differences between stretching and bending are clearly differentiated. Discussions are now improved.
- What are the uses of ethanol extracts?
Response: The rational of ethanol as an extracting solvent is now included at the beginning of the results and discussion.
- Conclusions presentation must be improved to highlight the novel aspects of the study.
Response: The novelty of this study is now included in the conclusion section.
- Methodology related to US parameters is minimal
Response: Further clarifications are now included.
Reviewer 3 Report
Comments and Suggestions for Authors
This study is conducted to structurally characterize micronized date-pits. The micronized date-pits were obtained by a combination of ultrasound treatment and alcoholic sedimentation. The authors used several techniques such as SEM, TEM, FTIR etc. to assess the output.
However, the rationale of this study is unclear. Ultrasound treatment was claimed to degrade micronized particles, the underlying mechanism was not discussed. The discussion section is very minimal with poor level of obtained results. FTIR Table and figures are redundant.
Six types of micronized date-pits powder with varying particle size and characteristics were obtained. However, its application in food products was not clearly presented.
In literature, several others have used date-pits for various applications and have processed date-pits with cost analysis and efficacy over conventional process.
Based on the above observation, the article may be rejected or revise completely.
Author Response
Referee 3
However, the rationale of this study is unclear. Ultrasound treatment was claimed to degrade micronized particles, the underlying mechanism was not discussed. The discussion section is very minimal with poor level of obtained results. FTIR Table and figures are redundant.
Response: Clearer rational of this study is now included. The mechanisms of micronized fractionation are now included for alcoholic sedimentation and ultrasound treatment. The discussion is now improved. FTIR tables are deleted and only FTIR spectra are retained as graphs.
Six types of micronized date-pits powder with varying particle size and characteristics were obtained. However, its application in food products was not clearly presented.
Response: The applications for the extracted date-pits are now included based on their properties.
In literature, several others have used date-pits for various applications and have processed date-pits with cost analysis and efficacy over conventional process.
Response: Different other extraction methods are now included.
Based on the above observation, the article may be rejected or revised completely.
Response: We have attempted to improve the content and clarification of the paper and hope that it will be acceptable now.